# Research on CdSe/ZnS Quantum Dots-Doped Polymer Fibers and Their Gain Characteristics

**DOI:** 10.3390/nano14171463

**Published:** 2024-09-09

**Authors:** Xuefeng Peng, Zhijian Wu, Yang Ding

**Affiliations:** 1College of Science & Technology, Ningbo University, Ningbo 315211, China; 2111082180@nbu.edu.cn (Z.W.); dingyang@nbu.edu.cn (Y.D.); 2Faculaty of Electrical Engineering and Computer Science, Ningbo University, Ningbo 315211, China

**Keywords:** Cdse/ZnS, quantum dots, polymer fiber, on-off gain, melt extrusion method

## Abstract

Polymer fibers are considered ideal transmission media for all-optical networks, but their high intrinsic loss significantly limits their practical use. Quantum dot-doped polymer fiber amplifiers are emerging as a promising solution to this issue and are becoming a significant focus of research in both academia and industry. Based on the properties of CdSe/ZnS quantum dots and PMMA material, this study experimentally explores three fabrication methods for CdSe/ZnS quantum dots-doped PMMA fibers: hollow fiber filling, melt-drawing, and melt extrusion. The advantages and disadvantages of each method and key issues in fiber fabrication are analyzed. Utilizing the CdSe/ZnS quantum dots-doped PMMA fibers that were fabricated, we theoretically analyzed the key factors affecting gain performance, including fiber length, quantum dots doping concentration, and signal light intensity. Under the conditions of 1.5 W power and 445 nm laser pumping, a maximum on-off gain of 16.2 dB was experimentally achieved at 635 nm. Additionally, using a white light LED as the signal source, a broadband on-off gain with a bandwidth exceeding 70 nm and a maximum gain of 12.4 dB was observed in the 580–650 nm range. This research will contribute to the development of quantum dots-doped fiber devices and broadband optical communication technology, providing more efficient solutions for future optical communication networks.

## 1. Introduction

With the increasing popularity of 4K TVs, high-definition monitoring, virtual reality devices, and smart home devices, the demand for broadband and high-speed communication networks has been growing in recent years [1]. All-optical networks are considered an effective means to enhance the bandwidth and data transmission speed for end-users [2]. Traditional fused silica fibers, with their small core diameter, poor bend resistance, and high coupling precision requirements, incur high costs due to the need for numerous connectors in communication networks. In contrast, polymer optical fibers (POFs), characterized by their larger core diameters (about 0.3–3 mm), offer notable advantages including enhanced flexibility, superior bend resistance, and cost-effectiveness, which collectively render them an attractive choice as the primary transmission medium in local area networks (LANs).

Researchers have been investigating the application of POFs in LANs for an extended period [3,4]. However, a significant challenge with POFs is their high attenuation. For instance, polymethyl methacrylate (PMMA) fibers, the common POF, exhibit an attenuation of approximately 100 dB/km [4]. This high loss restricts signal transmission distances to around 100 m, thereby presenting a limitation for deployment in expansive structures such as skyscrapers or sizable corporate complexes.

The high attenuation observed in PMMA fibers might be primarily attributed to the strong absorption properties of the material itself, specifically related to the C-H bond vibrations. To mitigate this issue, researchers have substituted the C-H bonds with C-D bonds using deuterated PMMA (PMMA-d8) or with C-Y bonds using Cyclic Transparent Optical Polymers (CYTOP), effectively reducing the attenuation to as low as 20 dB/km [5]. Despite the significant reduction in attenuation, these materials also substantially increase the cost of the fibers, thereby limiting their practical application. Innovative structural designs, such as hollow-core structures with photonic bandgap cladding, have shown potential in reducing attenuation by an order of magnitude [6]. However, high fabrication costs and integration challenges with existing communication systems continue to impede their large-scale deployment. Thus, for POFs to achieve widespread practical application in LANs, the introduction of polymer optical fiber amplifiers (POFAs) becomes necessary to counteract the high attenuation. The low-loss windows of POFs predominantly reside in the visible light spectrum. The two main categories of gain media that have been developed in POFAs are rare-earth chelates and organic dyes.

Rare-earth chelates such as Eu(DBM)_3_Phen, Eu(TTA)_3_Phen, and Sm(HFA)_4_Net_4_ have shown significant potential in the development of POFAs. For instance, Liang et al. [7] incorporated Eu(DBM)_3_Phen into PMMA fibers and achieved a gain of 5.7 dB at 613 nm when pumped with a 355 nm YAG laser. Hu et al. [8] embedded Eu(TTA)_3_Phen into liquid-core fibers, obtaining a gain of 8.2 dB at 615 nm with a 203 mW, 355 nm laser pump. Huang et al. [9] doped Sm(HFA)_4_Net_4_ into SU8 polymer material, achieving a gain of 7.4 dB at 645 nm under a 250 mW, 351 nm laser pump. However, the luminescence properties of these rare-earth chelates are significantly influenced by the organic ligands attached to them, leading to low quantum efficiency and high population inversion requirements, which inherently limit the achievable gain.

Organic dyes represent another extensively studied class of gain media for POFAs. Tagya et al. [10] demonstrated a gain of 33 dB at 580 nm by doping Rhodamine B into POFs and pumping with a 532 nm laser. Similarly, Mandamparambil et al. [11] incorporated Rhodamine 6G into POFs, achieving an 18 dB gain under 6 mJ pulse energy laser pumping. Sheeba et al. [12] mixed Rhodamine B and Rhodamine 6G in PMMA fibers, resulting in a 22.3 dB gain with a 60 nm bandwidth under Nd:YAG pulse laser excitation. Additionally, Arrue et al. [13] created graded-index fibers by doping Rhodamine 6G into PMMA, achieving broadband amplified spontaneous emission under side-pumping conditions. Despite these promising results, organic dyes are characterized by low stability and susceptibility to thermal bleaching, which limits their operational stability and longevity. Moreover, organic dyes exhibit fixed absorption and emission spectra. For example, Rhodamine B-doped PMMA POF amplifiers only cover the low-loss window around 580 nm, leaving the 650 nm window of PMMA fibers uncovered and thus restricting the gain bandwidth.

In recent years, quantum dots (QDs) have garnered significant attention due to their unique quantum size effect, dielectric confinement effect, and surface effect. Studies have shown that core–shell structure CdSe/ZnS QDs, whether deposited as films or embedded in waveguides [14,15,16,17], exhibit excellent gain and amplification characteristics, with emission peaks covering the low-loss windows of common POFs. Extensive studies have explored the integration of CdSe/ZnS QDs within POFs utilizing a diverse array of host materials, encompassing PMMA, polyethylene (PE), and ultraviolet (UV) curable adhesives [18,19,20]. Moreover, studies have shown that CdSe/ZnS QDs exhibit a pronounced resilience against photobleaching in comparison to traditional organic dyes [21,22]. Moreover, the low fabrication temperature of POFs (below 200 °C) does not adversely affect the luminescent properties of CdSe/ZnS QDs. These makes them suitable as gain media and present a promising alternative to traditional organic dyes and rare-earth chelates in POFAs. It is imperative to highlight that any wavelength shorter than the emission wavelength of QDs can serve as the pump, effectively avoiding the high costs associated with fixed excitation wavelengths in other gain media. 

This study leverages the solubility of CdSe/ZnS QDs and PMMA in organic solvents, along with the low drawing temperature of PMMA fibers, to experimentally fabricate CdSe/ZnS QDs-doped PMMA fibers using the hollow fiber filling method, the melt-drawing method, and the melt extrusion method. Each method’s merits, drawbacks, and critical issues pertinent to fiber preparation were analyzed. Based on the fabricated fibers, we experimentally achieved broadband high-gain optical amplification with a gain bandwidth of 70 nm (spanning 580–650 nm) and a maximum on-off gain of 16.2 dB. This research effectively promotes the development of QDs-doped fiber lasers and amplifiers and holds promise for advancing full-band broadband optical amplification technology and the broadband optical communication industry.

## 2. Experimental Section

### 2.1. Material Characterization

The CdSe/ZnS QDs were provided by Shanghai Institute of Technical Physics, Chinese Academy of Sciences. The PMMA powder, with a molecular weight of 12,000, was purchased from a commercial supplier (Macklin Biochemical, Shanghai, China). All materials employed in this research were of analytical grade. The morphology and absorption-emission spectra of the QDs are illustrated in Figure 1. Transmission electron microscopy (TEM) images, captured using a Talos F200X microscope (Thermo Fisher Scientific, Waltham, MA, USA), indicated an average QDs diameter of 4.46 ± 0.64 nm. Using the energy-dispersive spectroscopy (EDS) attachment (Thermo Fisher Scientific, Waltham, MA, USA) on the Talos F200X, high-angle annular dark field (HAADF) imaging was performed. The subsequent analysis revealed that the elemental composition of Cd, Se, Zn, and S in CdSe/ZnS QDs is approximately 39.42%, 6.86%, 11.66%, and 42.06%, respectively. The absorption spectrum of the QDs was recorded using a Cary 300 UV-Vis.

### 2.2. Fabrication of QDs-Doped PMMA Fibers

The fabrication of QDs-doped POFs represents a critical initial step in developing optical fiber amplifiers. To determine the most suitable fabrication techniques and parameters, we investigated three methods for producing polymer optical fibers utilizing CdSe/ZnS QDs and PMMA: the hollow fiber filling method, the melt-drawing method, and the melt extrusion method. We conducted a comprehensive analysis of the key factors that affect the optical fiber performance parameters, aiming to optimize the fabrication process for enhanced fiber quality and functionality.

#### 2.2.1. Hollow Fiber Filling Method

The ester groups in PMMA interact readily with polar solvents, allowing PMMA to dissolve in solvents such as chloroform when heated and stirred. This property enables the preparation of a QDs-doped polymer colloid by dissolving CdSe/ZnS QDs and PMMA in chloroform. The resulting colloid can then be injected into a hollow fiber using a pressure differential filling technique to produce QDs-doped PMMA fibers. Due to the toxicity of chloroform and its susceptibility to oxidation in air, which can produce corrosive hydrogen chloride and highly toxic phosgene, the fabrication process must be carried out in a fume hood to ensure safe handling and minimize exposure risks.

The fabrication process was similar to that outlined by Whittaker et al. [23], as illustrated in Figure 2, and involves several key steps. Initially, chloroform was incrementally added to PMMA powder, and the mixture is subjected to ultrasonic agitation in a controlled heated environment to create a PMMA colloid. CdSe/ZnS QDs were then dissolved in toluene, and this solution was combined with the PMMA colloid, again using ultrasonic agitation to ensure homogenous dispersion of the QDs throughout the colloid. The resulting colloid was then placed in a rotary evaporator, where the temperature is maintained at 111 °C (slightly above toluene’s boiling point of 110.6 °C) to effectively remove toluene. Following solvent removal, the colloid underwent a degassing process in a vacuum chamber for 20 min to eliminate any entrapped air bubbles, which could otherwise compromise the optical clarity and performance of the fiber. Finally, the degassed colloid was introduced into a hollow fiber using a vacuum-induced pressure differential. Due to the colloid’s inherent viscosity and limited flow characteristics, meticulous control over the filling rate and the pressure differential is essential. This precision ensures consistent and uniform filling of the fiber core, thereby maintaining the desired optical and mechanical properties of the QDs-doped PMMA fibers.

#### 2.2.2. Melt-Drawing Method

PMMA exhibits a melting point of approximately 105 °C and a flow temperature around 160 °C. At moderately elevated temperatures (approximately 200 °C), the QDs-doped PMMA material transitions into a molten state. This molten material can be drawn into fibers using a pointed implement [24]. This technique allows for the controlled formation of fibers with specific diameters and properties. A detailed schematic of the melt-drawing method utilized for fabricating QDs-doped POFs is presented in Figure 3.

The detailed process for fabricating optical fibers using the melt-drawing method is as follows: Begin by setting the temperature-controlled heating plate to a suitable temperature range (200–230 °C) and allow it to reach the set temperature. Thoroughly mix the QDs with PMMA powder, then place the mixture on the preheated stage. Stir continuously until the mixture is completely molten. Insert a conical pointed tool into the molten material and draw the fiber at a uniform speed. After drawing, place the fiber in a drying oven to cool and solidify. 

#### 2.2.3. Melt Extrusion Method

In the industrial production of PMMA POFs, the melt extrusion method is commonly used. This method involves producing PMMA via bulk polymerization, followed by extrusion, drawing, fiber winding, and sheathing, forming a continuous production line for PMMA POFs. However, conventional extrusion equipment faces challenges in precisely controlling the concentration and distribution of functional materials like quantum dots during the fiber production process. To overcome this limitation, we employed a custom-built desktop-scale POF extruder to fabricate QDs-doped PMMA fibers. The equipment setup, illustrated in Figure 4, is arranged from right to left as follows:

Extruder: This component heats the quantum dot-doped PMMA to a molten state and forces it through a die to shape it into a fiber.

Cooling Trough: Immediately after extrusion, the hot fiber passes through a cooling trough where it solidifies and stabilizes its structure.

Fiber Dimension Detector: An inline measurement device that continuously monitors the fiber’s dimensions, ensuring consistency in diameter and other critical parameters.

Fiber Winder: Once cooled and set, the fiber is wound onto a spool by the winder for collection and further processing.

The process for fabricating POFs using the melt extrusion method involves several key steps. First, QDs were thoroughly mixed with PMMA powder, and the mixture was preheated and dried at approximately 80 °C to reduce moisture content, which can impact fiber quality. The extruder is then preheated, setting the optimal temperature for PMMA extrusion at around 196 °C, including heating the extrusion head. Once the materials reach the molten state, they are extruded through the extrusion head by starting the extruder motor. Finally, the fiber is wound onto spools using a fiber winder, with the winding speed set at approximately 0.05 m/s in this experiment, adjusted along with the extruder temperature to achieve optimal synchronization and stable fiber output. 

### 2.3. Spectral and Optical Amplification Characteristics Testing

To evaluate the gain characteristics of CdSe/ZnS QDs-doped PMMA fibers, we established a testing system for both spectral and gain properties, as depicted in Figure 5. The signal light had a central wavelength of 635 nm and a maximum output power of 30 mW (Infrared Laser Tech., Shenzhen, China). The pump light was a semiconductor fiber laser (Dragonback Laser Tech., Beijing, China), with a central wavelength of 445 nm and a maximum output power of 1.5 W. Both the signal and pump lights were combined using a wavelength-division multiplexer (HJGTEK Tech., Shenzhen, China) and then guided through a multimode silica optical fiber (core/cladding dimensions 62.5/125 μm) into the CdSe/ZnS QDs-doped PMMA fiber. The output signal was collected via another multimode silica optical fiber (core/cladding dimensions 62.5/125 μm) and directed into a spectrometer (Joule Tech., Hangzhou, China). The wavelength range of the spectrometer is 390–1020 nm. The QDs-doped fiber used an SMA905 connector (NLG Tech., Shenzhen, China), which was connected to the silica optical fiber using an SMA905 to FC adapter.

The emission peak wavelength of the CdSe/ZnS QDs was centered at 608.7 nm. After passing through approximately 10 cm of the QDs-doped fiber, the peak wavelength red-shifted to 625.2 nm. This red-shift brought the emission wavelength closer to the signal light wavelength of 635 nm, thereby facilitating higher signal light gain.

## 3. Theoretical Model

As shown in Figure 1, CdSe/ZnS QDs exhibit a single-peak emission, indicating that the radiative process occurs exclusively between two energy levels. The absorption and emission processes of CdSe/ZnS QDs can be modeled by a three-level system, as depicted in Figure 6. In this schematic, energy level 1 represents the ground state, while energy level 2 includes two fine structure states, corresponding to the first absorption peak and emission peak of CdSe/ZnS QDs. Energy level 3 comprises a set of energy states, associated with the continuous absorption observed in the short-wavelength region of Figure 1.

Upon excitation with short-wavelength pump light, the QDs absorb energy and are excited to energy levels 2 and 3, as illustrated by the dashed transitions in Figure 5. Particles in energy level 2 return to the ground state via both stimulated and spontaneous emission. Due to parity selection rules, particles in energy level 3 cannot directly transition to the ground state through radiative processes. Instead, they undergo a non-radiative transition to energy level 2 with a probability denoted as A_32_, followed by a radiative transition back to the ground state. The transition probability from energy level 3 to energy level 2 is significantly high, classified as an intraband transition. As a result, particles in energy level 3 rapidly transition to energy level 2, thereby allowing the three-level system of the CdSe/ZnS QDs to be effectively approximated by a two-level system [25].

The dynamics of the pumping light power (*P_P_*), emitted light power (*P)*, and the molecule population density in the excited state (*N_2_*) as functions of time (*t*) and position along the fiber (*z*) can be analyzed using a set of rate equations similar to those employed for rare-earth-doped fibers. Notably, the emission spectra of CdSe/ZnS QDs-doped POFs can be significantly tuned by adjusting the fiber length, leveraging the considerable overlap between the absorption and emission cross-sections of CdSe/ZnS QDs. This necessitates considering wavelength (*λ*) as an independent variable, enabling computational simulations of spectra and their evolution with fiber parameters, such as QDs doping concentration and fiber length.

To incorporate this wavelength dependence, the absorption and photoluminescence spectra are divided into discrete subintervals centered at specific wavelengths *λ_k_*. Consequently, both the absorption and emission cross-sections are wavelength-dependent.

Assuming end-pumping of the CdSe/ZnS QDs-doped POFs from a specific position with a pump laser of a particular wavelength, the rate equations and the light power propagation equation can be expressed as follows [26]:(1)$$\begin{array}{ll} \frac{\partial N_{2}(z,t)}{\partial t}= & -\frac{N_{2}(z,t)}{\tau }+\frac{\lambda_{P}\Gamma_{P}}{hcA_{core}}[\sigma_{a}(\lambda_{P})N_{1}(z,t)-\sigma_{e}(\lambda_{P})N_{2}(z,t)]P_{P}(z,t)\\ & +\frac{\Gamma_{e}}{hcA_{core}}\sum_{k=1}^{K}\lambda_{k}[\sigma_{a}(\lambda_{k})N_{1}(z,t)-\sigma_{e}(\lambda_{k})N_{2}(z,t)]P(z,t,\lambda_{k}) \end{array}$$
(2)$$\begin{array}{ll} \frac{\partial P(z,t,\lambda_{k})}{\partial z}= & \Gamma_{e}[\sigma_{e}(\lambda_{k})N_{2}(z,t)-\sigma_{a}(\lambda_{k})N_{1}(z,t)]P(z,t,\lambda_{k})-\alpha (\lambda_{k})P(z,t,\lambda_{k})\\ & -\frac{1}{v_{e}}\frac{\partial P(z,t,\lambda_{k})}{\partial t}+\frac{N_{2}(z,t)}{\tau }h(\frac{c}{\lambda_{k}})\sigma_{e-sp}(\lambda_{k})\beta A_{core}\;\;\;\;\;k=1,\ldots K \end{array}$$
(3)$$\frac{\partial P_{P}(z,t)}{\partial z}=-\Gamma_{P}[\sigma_{a}(\lambda_{P})N_{1}(z,t)-\sigma_{e}(\lambda_{P})N_{2}(z,t)]P_{P}(z,t)-\alpha (\lambda_{P})P_{P}(z,t)-\frac{1}{v_{P}}\frac{\partial P_{P}(z,t)}{\partial t}$$

*N_2_*(*z, t*) and *P_P_*(*z, t*) represent the number of particles in the upper energy level and the pump light intensity at position *z* and time *t*, respectively. *P*(*z, t, λ_k_*) denotes the signal light intensity at position *z*, time *t*, and wavelength *λ_k_*. The last term in Equation (1) corresponds to the “reabsorption” effect [26]. Finding an analytical solution to these partial differential equations is challenging, necessitating the use of finite difference methods to obtain a numerical solution. To proceed, we first establish a two-dimensional matrix *N_2_* and *P_P_*, and a three-dimensional matrix *P*(*z, t, k*), where *k* denotes different wavelength segments. To analyze the light amplification at each wavelength, we divide the entire spectrum into *k* equal or unequal segments, each corresponding to a central wavelength *λ_k_*. Considering the “reabsorption” effect, the absorption and emission cross-sections are also divided into *k* segments, with each segment having the same absorption and emission cross-sections, denoted as *σ_a_*(*λ_k_*) and *σ_e_*(*λ_k_*), respectively.

By incorporating the initial and boundary conditions for the CdSe/ZnS QDs-doped POFAs and iterating through the finite difference method, the final output power can be obtained. The system’s final gain coefficient can be expressed as $G=10\mathrm{lg}(P_{out}/P_{in})$. The parameters can be calculated as the methods mentioned in the literature [26].

In the theoretical model described, assuming a high pump light intensity *P_P_*, we can analyze different scenarios based on the initial conditions at *z* = 0 and *t* = 0. If the signal light intensity *P* = 0, the model calculates the amplified spontaneous emission (ASE). Conversely, if an initial signal light intensity *P* ≠ 0 is present, the model calculates the amplification of the signal light. Under the assumption of a low pump light intensity, with P = 0 at z = 0 and t = 0, the emission spectrum of the CdSe/ZnS QDs-doped POFs without optical amplification. Thus, this theoretical model is applicable to three distinct scenarios: calculating the output emission spectrum without amplification, determining the Amplified Spontaneous Emission (ASE) in the absence of initial signal light, and assessing the amplification of signal light when an initial signal is present.

## 4. Results and Discussion

### 4.1. Issues in the Fabrication of QDs-Doped PMMA Fibers

#### 4.1.1. Advantages and Disadvantages of Hollow Fiber Filling Method

The hollow fiber filling method is relatively easy to implement, producing fibers with good uniformity and allowing precise control over the doping concentration of QDs. This method is effective for small-scale fiber production. However, there are several drawbacks associated with this technique:Safety Concerns: The fabrication process involves the use of toxic solvents such as toluene and chloroform, which pose safety risks.Length Limitation: The length of the fabricated fibers is constrained by the length of the hollow fiber, making this method unsuitable for large-scale fiber production.Bubble Formation: During the mixing of PMMA powder with chloroform, numerous air bubbles can form, which will significantly impact the fiber’s performance if not effectively removed. Additionally, during the curing of the QDs-doped PMMA colloid, internal stresses can arise due to differences in materials, potentially leading to bubble formation between the fiber core and cladding (as shown in Figure 7), further affecting the optical fiber’s performance.

#### 4.1.2. Advantages and Disadvantages of Melt-Drawing Method

The melt-drawing method is characterized by its simplicity and minimal equipment requirements, making it a viable approach for exploratory experiments. However, this method has several limitations:Limited Fiber Length: The fibers produced using this method are typically short, limiting its suitability for large-scale production.Control Over Fiber Diameter and Structure: Precise control over the fiber’s diameter and structural integrity is difficult to achieve during the drawing process. The cross-sectional shape of the fibers often deviates from the ideal circular form and may include voids (as shown in Figure 8). These imperfections can adversely affect the optical performance of the fibers and present challenges in the fabrication of subsequent optical fiber jumpers.

#### 4.1.3. Advantages and Disadvantages of Melt Extrusion Method

The melt extrusion method provides high precision in fiber fabrication, enabling accurate control over QDs doping concentration and fiber dimensions, which is advantageous for large-scale production. However, the fiber’s performance is highly sensitive to the heating temperature and drawing speed, necessitating precise coordination of these parameters to maintain optimal fiber quality.

In our current setup, while fiber dimensions can be monitored during the drawing process, there is no real-time feedback system to dynamically adjust these parameters. This absence of real-time adjustment capability affects the uniformity of the fiber dimensions, suggesting a need for further refinement. An example of a fiber produced using the extrusion method is shown in Figure 9.

#### 4.1.4. Choose of the Suitable Fabrication Method

In many prior studies, researchers employed the hollow fiber filling method to prepare liquid-core and polymer optical fibers and utilized the melt-drawing method for fabricating polymer and tapered fibers. Therefore, as a comparative study, this work explores the preparation of CdSe/ZnS quantum dot-doped PMMA fibers using the three methods mentioned above and analyzes the advantages and disadvantages of each preparation technique. However, in polymer optical fiber amplifiers (POFAs), the polymer fibers are typically required to have minimal defects, a good structure, ease of splicing, and a long fiber length. The hollow fiber filling method is prone to bubble formation, which adversely affects fiber performance. The melt-drawing method struggles to achieve standard cylindrical cross-section fibers, leading to higher connection and transmission losses. In contrast, the melt extrusion method produces fibers with a standardized structure, facilitates easy splicing, and imposes virtually no limitations on fiber length. Therefore, we choose the melt extrusion method for the fabrication of CdSe/ZnS quantum dot-doped PMMA optical fibers in our work.

### 4.2. Theoretical Analysis of Spectral and Amplification Performance

Based on the theoretical model discussed above, we calculated the gain spectra of CdSe/ZnS QDs-doped PMMA fiber amplifiers and analyzed the relationships between the gain and the length of the doped fiber, the concentration of QDs doping, and the signal light intensity.

Figure 10 illustrates the variation of gain with the length of QDs-doped fiber, where the CdSe/ZnS doping concentration was 2 ppm and the fiber diameter was 250 μm. The fiber was pumped with a pulsed laser having a pulse width of 20 ps, a frequency of 10 Hz, and a pump energy density ranging from 0.1 to 9 mJ/cm^2^. As depicted in Figure 10a,c, the gain increases with the pump energy density, reaching up to 13 dB at a pulse energy density of 9 mJ/cm^2^. Furthermore, under specific pump light intensity conditions, the gain initially increases and then decreases with the length of the doped fiber, forming a characteristic curve (Figure 10c). This single-peaked phenomenon, similar to that observed in conventional fibers, results from the competition between gain and loss along the fiber.

Figure 10b also presents the fiber length corresponding to the maximum gain under different pump light intensities, indicating the optimal fiber length. When the pump intensity is relatively low (0.1–2 mJ/cm^2^), the optimal fiber length increases linearly from 3 cm to 8 cm. As the pump light intensity continues to increase, the growth rate of the optimal QDs-doped fiber length slows down, stabilizing at around 10–11 cm. This indicates that the fiber length does not need to be excessively long, as a relatively short fiber length (around 10 cm) can achieve a gain of over 10 dB.

Figure 11 illustrates the relationship between gain, doped fiber length, and QDs doping concentration under fixed pump and signal light intensities. When the QDs doping concentration exceeds the optimal number that can be effectively excited by the pump light, the signal gain decreases rapidly. Therefore, the doping concentration should be carefully optimized, as excessive doping not only increases costs but also degrades the amplifier’s gain performance.

The relationship between gain and signal light intensity is shown in Figure 12. As depicted, the signal light intensity exerts a limited impact on the gain. While a slight reduction in gain is observed with increasing signal light intensity under various pump intensities, the overall variation is minimal. For example, within a signal light intensity range of 2–10 dBm and a pump energy density of 40 mJ/cm^2^, the maximum gain reaches 16.5 dB, whereas the minimum gain is 15.3 dB. Consequently, the QDs-doped POFAs exhibit low sensitivity to signal light intensity and maintain nearly a flat gain over a broad range of signal light intensities.

Under the conditions of a CdSe/ZnS QDs doping concentration of 2 ppm, a doped fiber length of 10 cm, a signal light intensity of 3 dBm, and a pump energy density ranging from 5 to 60 mJ/cm^2^, the relationship between the gain spectra and wavelength is illustrated in Figure 13. As shown in the figure, with increasing pump intensity, the gain significantly rises from 8.4 dB to 21.1 dB. However, the full width at half maximum (FWHM) of the gain spectrum remains unchanged. This indicates that the gain bandwidth of the CdSe/ZnS QDs-doped PMMA fiber is fixed, allowing for a high gain over a broad wavelength range.

### 4.3. Experimental Measurement of On-Off Gain

To validate the feasibility of the QDs-doped POFAs, we experimentally measured the on-off gain characteristics of the CdSe/ZnS QDs-doped PMMA fiber amplifier under continuous laser pumping. The on-off gain was determined as the ratio of the output signal powers when the pump lasers were tuned on and off. The experimental setup used for these measurements is based on the configuration shown in Figure 5.

The experimental results illustrating the variation of on-off gain with pump power are presented in Figure 14. The on-off gain for three different lengths of doped fibers (8 cm, 10 cm, and 12 cm) exhibits similar trends. At relatively low pump power levels (approximately < 0.3 W), the gain increases nearly linearly with the pump power. However, when the pump power exceeds 0.5 W, the gain increment significantly slows down and approaches saturation. Consistent with the simulation results shown in Figure 9 and Figure 10, longer fiber lengths do not necessarily result in higher on-off gain. Under the conditions of a 2 ppm doping concentration, 1.5 W pump power, and 3 dBm signal power, the maximum on-off gain of 16.2 dB was achieved with an 8 cm long doped fiber. The gain performance is comparable to that of PbS or PbSe QDs-doped fiber amplifiers operating in the 1.55 µm wavelength band [27,28]. It is important to note that the central wavelength of the signal light is 635 nm, while the peak wavelength at the output end of the 8 cm long doped fiber, influenced by the “reabsorption effect” due to the overlap of absorption and emission spectra, is approximately 625 nm. This mismatch affects the on-off gain. Theoretically, if the system parameters are adjusted so that the signal light wavelength precisely aligns with the peak output spectrum wavelength, the on-off gain coefficient could be significantly increased.

Since the on-off gain is not entirely due to energy level transitions resulting from population inversion, the experimentally measured on-off gain of the signal light is significantly influenced by the signal light intensity, unlike the theoretically calculated net gain which is less affected by signal light intensity. As illustrated in Figure 15, the on-off gain curves for three different signal light intensities show an initial rapid increase followed by a plateau. Under the same pump power, lower signal light intensity results in higher on-off gain. For a fiber length of 8 cm and a doping concentration of 2 ppm, with signal light intensities of 3 dBm, 7 dBm, and 10 dBm, the maximum on-off gains at a pump power of 1.5 W are 12.1 dB, 7.8 dB, and 5.2 dB, respectively.

To further analyze the gain spectrum characteristics of the QDs-doped fiber amplifier and achieve broadband optical amplification, we used a white light LED with an output wavelength range of 420–700 nm as the signal light source. The light was coupled into the QDs-doped fiber through a focusing lens. The on-off gain spectra were experimentally measured under 15 different pump power conditions, ranging from 0.1 W to 1.5 W in 0.1 W increments, as shown in Figure 16. The results demonstrate that significant on-off gain was achieved across the QDs emission peak range of 580–650 nm, with a maximum on-off gain of 12.4 dB. The measured on-off gain was higher in the longer wavelength region (620–650 nm) compared to the shorter wavelength region (580–620 nm), which is primarily due to the higher signal light intensity in these wavelength bands. The on-off gain and gain bandwidth achieved in this study are comparable to or even exceed those of PbS/CdS core–shell QDs-doped NOA85 UV glue fiber amplifiers [29].

Although we successfully achieved on-off gain in CdSe/ZnS QDs-doped PMMA fibers using both 635 nm laser and white light LED as signal sources, net gain was not realized due to experimental limitations. Several factors constrain the achievement of net gain in QDs-doped POFs:

The highly degenerate valence and conduction bands of CdSe/ZnS QDs, with the conduction band being doubly degenerate and the valence band degenerate six-fold, lead to additional energy level splitting due to spin–orbit coupling [30,31]. Achieving population inversion requires the involvement of more than one exciton, which is essential for QDs optical amplification. Additionally, the multiexciton process involves nonradiative Auger recombination with a lifetime of approximately 2 to 5 ps [32]. Achieving net gain necessitates population inversion within this short timeframe, requiring femtosecond or picosecond laser pumping. Current optical amplification tests, conducted under continuous-wave laser pumping, fail to meet this requirement. Furthermore, while high-power ultrafast laser pumping is necessary for net gain, it poses the risk of irreversible damage to the QDs. Finally, QDs are also prone to oxidation when exposed to air for prolonged periods. Although the polymer matrix provides stable encapsulation, protecting the QDs from water and oxygen, the inherent loss of the polymer matrix is significantly higher than that of fused silica fibers, limiting the potential for net gain.

Future efforts will focus on refining the core–shell structures to minimize surface defect states and Auger recombination and optimizing the polymer components to reduce fiber background losses, aiming to achieve substantial optical gain under lower ultrafast laser pumping conditions.

## 5. Conclusions 

This paper investigates the feasibility and underlying mechanisms of broadband optical amplification in CdSe/ZnS QDs-doped POFs. Through theoretical analysis, the gain characteristics of the amplifier under high-power pulsed laser pumping are examined. Three fabrication methods for QDs-doped POFs are proposed: the hollow fiber filling method, the melt-drawing method, and the melt extrusion method. Each method’s advantages and disadvantages are thoroughly analyzed. CdSe/ZnS QDs-doped PMMA fibers were fabricated using the melt extrusion method, and their optical amplification properties were experimentally studied. The experimental results demonstrated that, with a 635 nm laser as the signal light, a maximum on-off gain of 16.2 dB was achieved under the conditions of a 2 ppm doping concentration, an 8 cm fiber length, 1.5 W pump power, and 3 dBm signal power. When a white light LED was used as the signal light, a broadband on-off gain was observed within the 580–650 nm range, with a peak gain of 12.4 dB. The factors contributing to the failure to achieve net gain in certain experiments were analyzed.

This research provides a viable pathway for developing tunable wavelength fiber lasers and broadband fiber amplifiers based on colloidal QDs, which might be helpful to contribute to the development of optical communication technologies and industries.

## Figures and Tables

**Figure 1 nanomaterials-14-01463-f001:**
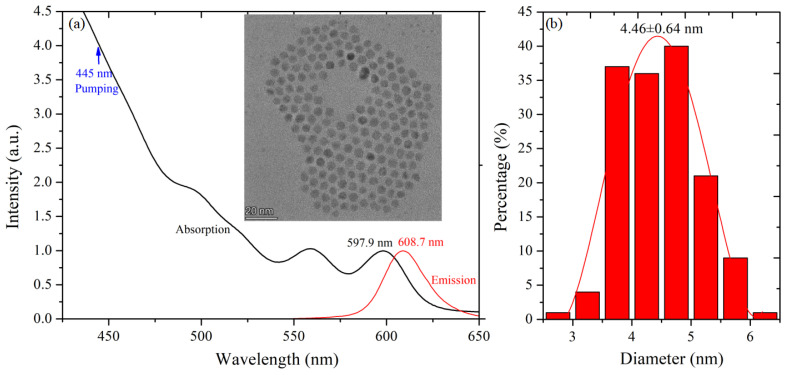
(**a**) Emission and absorption spectra and TEM image of CdSe/ZnS QDs; (**b**) the corresponding average QD diameter.

**Figure 2 nanomaterials-14-01463-f002:**
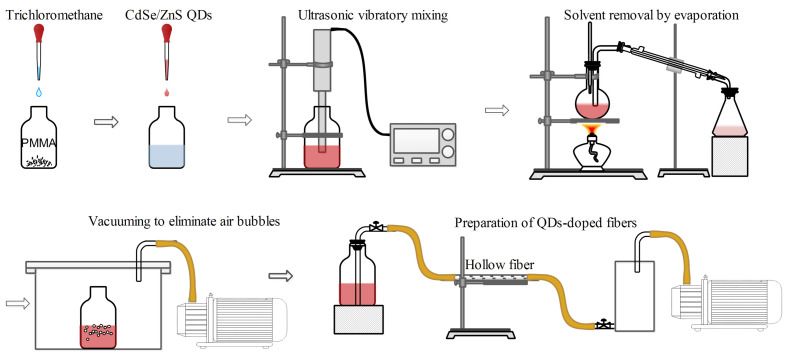
Schematic diagram of the experimental process for fabricating CdSe/ZnS QDs-doped PMMA fibers using the hollow fiber filling method.

**Figure 3 nanomaterials-14-01463-f003:**
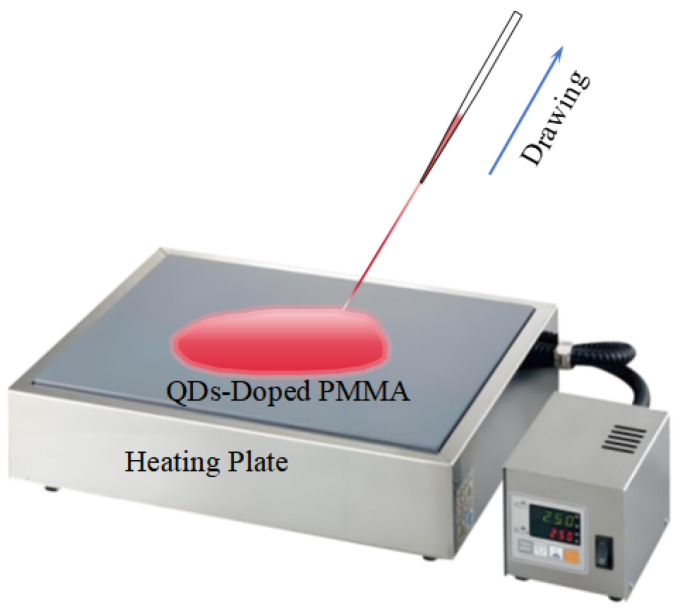
Schematic diagram of the melt-drawing method for fabricating QDs-doped POFs.

**Figure 4 nanomaterials-14-01463-f004:**
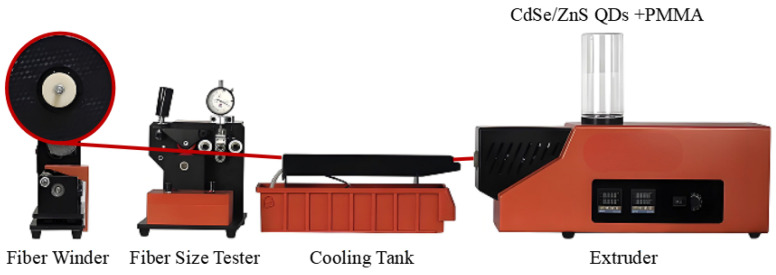
Schematic diagram of the melt extrusion method for fabricating QDs-doped POFs.

**Figure 5 nanomaterials-14-01463-f005:**
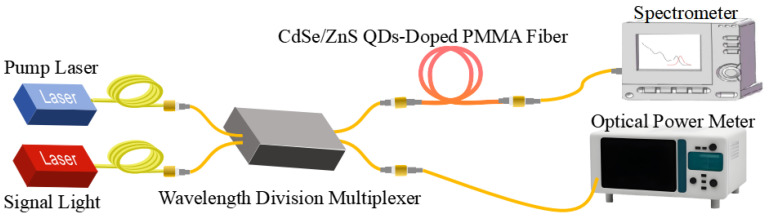
Schematic diagram of the optical amplification testing setup for CdSe/ZnS QDs-doped PMMA fibers.

**Figure 6 nanomaterials-14-01463-f006:**
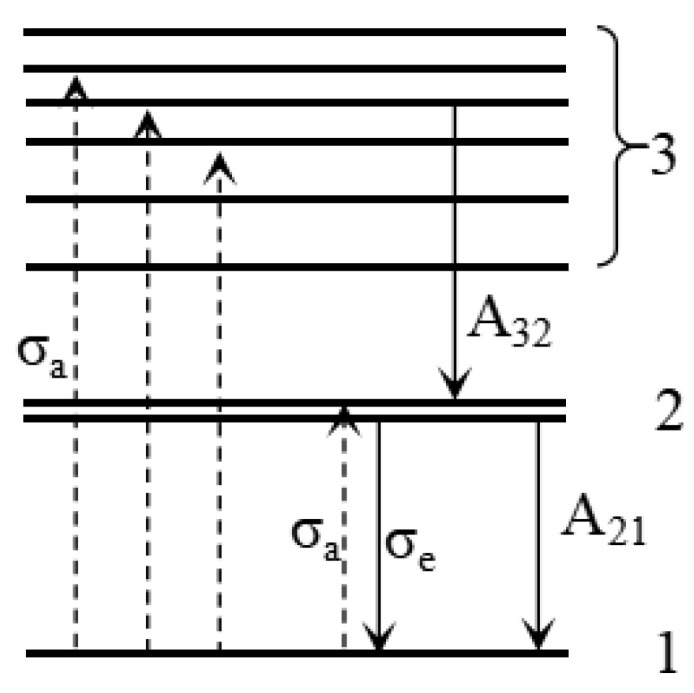
Energy levels of the CdSe/ZnS QDs.

**Figure 7 nanomaterials-14-01463-f007:**
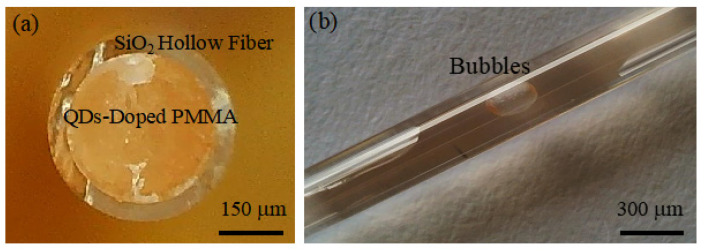
(**a**) CdSe/ZnS QDs-doped PMMA fiber fabricated using the hollow fiber filling method; (**b**) bubbles in the QDs-doped fiber.

**Figure 8 nanomaterials-14-01463-f008:**
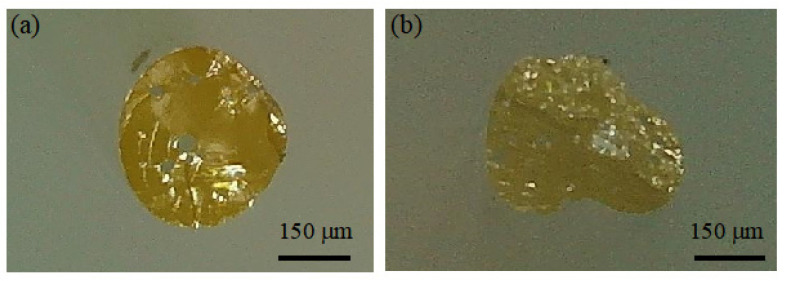
Cross-sectional image of QDs-doped fibers fabricated using the melt-drawing method: (**a**) there are voids within the optical fiber; (**b**) the POF exhibits an asymmetric structure.

**Figure 9 nanomaterials-14-01463-f009:**
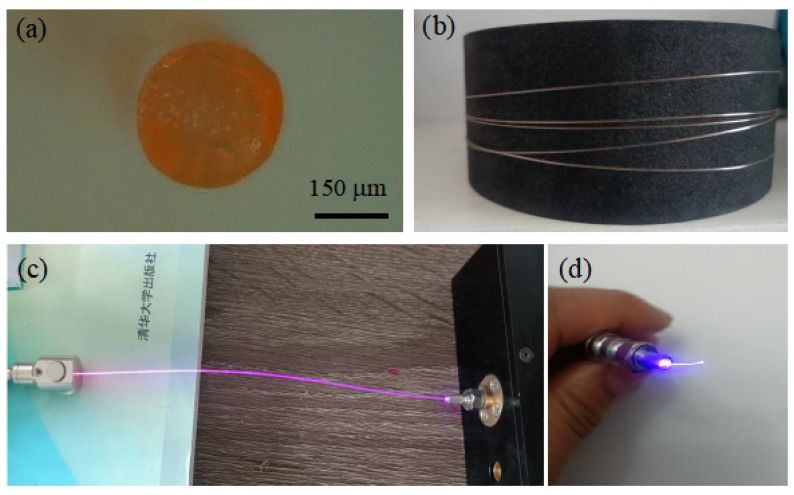
CdSe/ZnS QDs-doped PMMA fiber fabricated using the melt extrusion method: (**a**) microscopic image of the fiber end face, (**b**) a single long fiber strand, (**c**) QDs-doped PMMA fiber under 445 nm laser pump, and (**d**) a section of the fabricated QDs-doped PMMA fiber jumper.

**Figure 10 nanomaterials-14-01463-f010:**
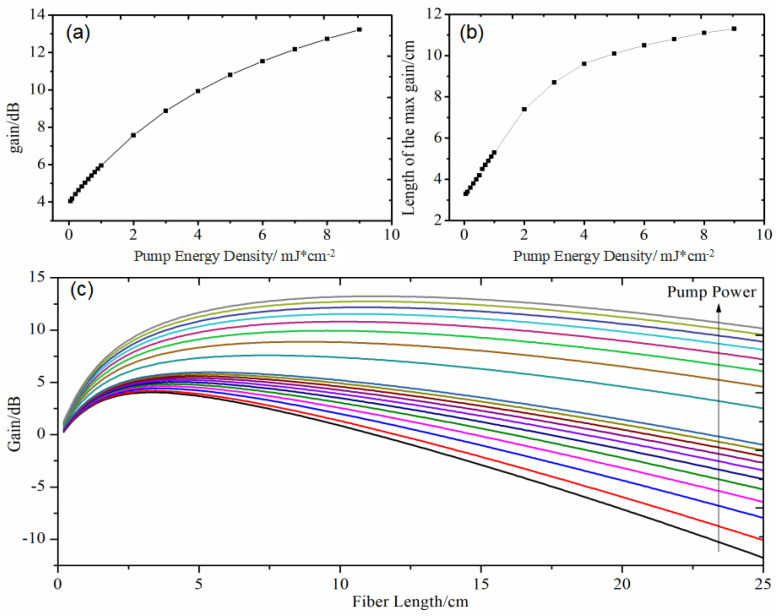
Maximum gain (**a**), optimal doped fiber length (**b**), and gain spectrum (**c**) as a function of quantum dot-doped polymer fiber length.

**Figure 11 nanomaterials-14-01463-f011:**
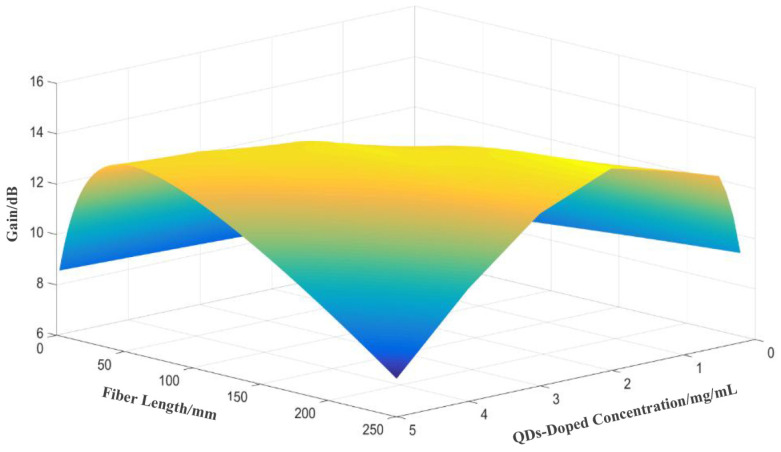
Relationship between gain, doped fiber length, and QDs doping concentration.

**Figure 12 nanomaterials-14-01463-f012:**
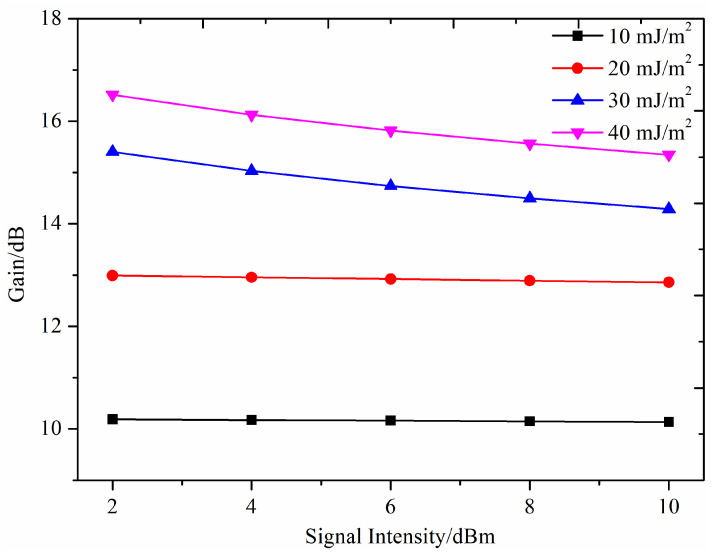
Variation of gain with signal light intensity.

**Figure 13 nanomaterials-14-01463-f013:**
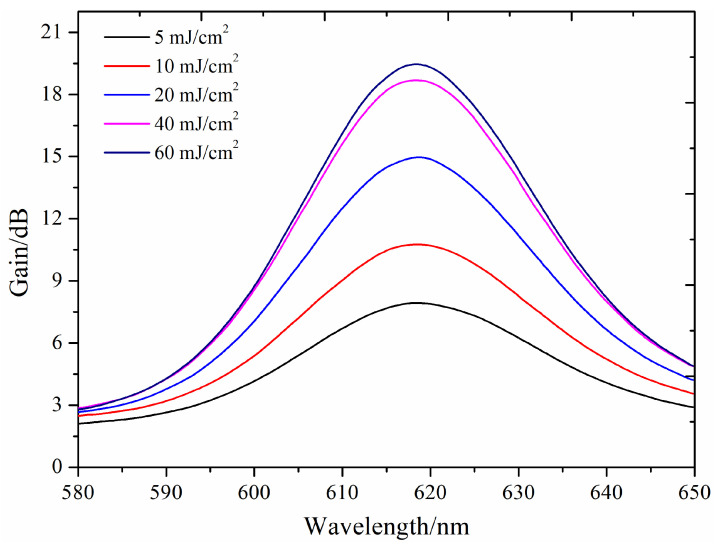
Gain spectra of CdSe/ZnS QDs-doped PMMA fiber amplifier at different pump energy densities.

**Figure 14 nanomaterials-14-01463-f014:**
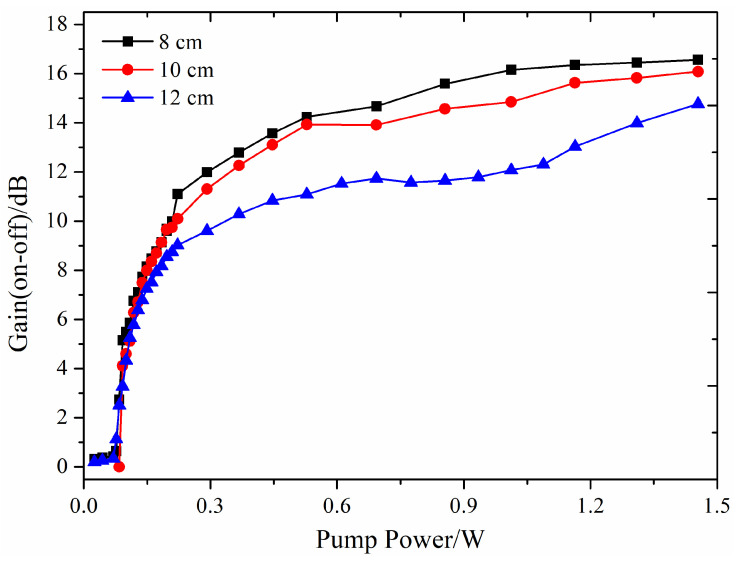
Experimental on-off gain as a function of pump power.

**Figure 15 nanomaterials-14-01463-f015:**
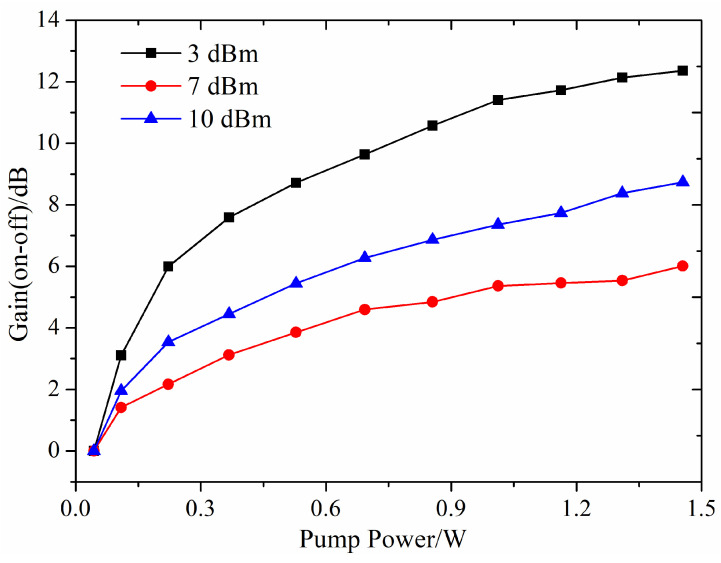
Variation of on-off gain with pump power at different signal light intensities.

**Figure 16 nanomaterials-14-01463-f016:**
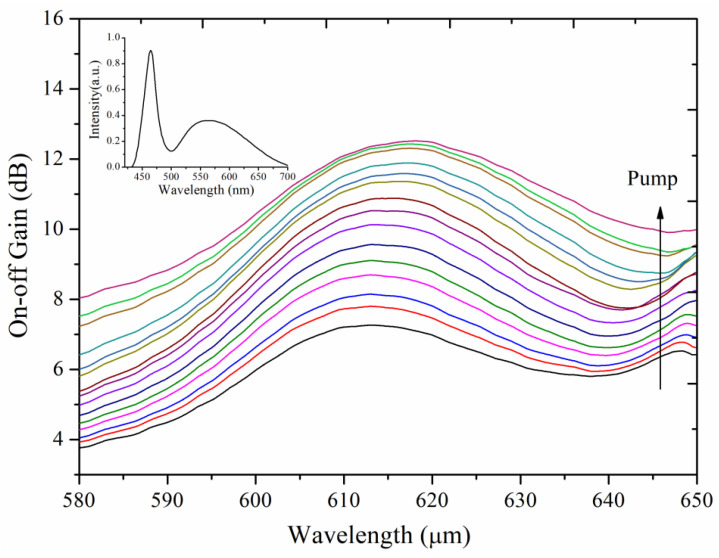
Variation of on-off gain spectra with pump power, inset showing the LED light spectrum.

## Data Availability

The data presented in this study are available on request from the corresponding author. The data are not publicly available due to further study.

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
