# Peer review of "Research on CdSe/ZnS Quantum Dots-Doped Polymer Fibers and Their Gain Characteristics"

_nanomaterials, 2024, doi:10.3390/nano14171463_

Round 1
Reviewer 1 Report
Comments and Suggestions for Authors
In this manuscript, the authors report on the fabrication of CdSe/ZnS QDs/PMMA composite fibers and their optical properties, especially broadband optical amplification. This study is very interesting as a basic investigation study to expand the application possibilities of inorganic QDs/polymer composite optical materials.
The authors' manuscript is well structured, interesting, and suggestive, but contains the following minor issues that could be improved.
1) There is information on the particle size of CdSe/ZnS, but not on the grade. What is the ratio of these elements in the “CdSe/ZnS” used in this study?
2) The authors should provide a clear explanation for the absence of a description of the PMMA grade, molecular weight, or purity. If these details are not necessary for this study, a comprehensive justification from the authors would be greatly appreciated.
3) Are the CdSe/ZnS QDs uniformly dispersed in the composite material? How did the authors confirm this? If there is anything that can be considered from the optical properties, it should be described and discussed.
4) There are many previous studies on the optical properties of CdSe/ZnS/polymer composites, and the authors' study should be compared with these studies to discuss what features are unique to these materials.
5) "2.2.1. hollow fiber filling method": The beginning of the sentence should be capitalized.
Author Response
Comments 1: In There is information on the particle size of CdSe/ZnS, but not on the grade. What is the ratio of these elements in the “CdSe/ZnS” used in this study?
Response:
Thanks for the reviewer’s comment. All materials used in this study were of analytical grade. Since the elemental ratio in the CdSe/ZnS QDs was not analyzed in the initial draft, we conducted additional experiments to determine the elemental composition following the review feedback. Using the energy-dispersive spectroscopy (EDS) attachment on the Talos F200x high-resolution field emission transmission electron microscope (HRTEM), high-angle annular dark field (HAADF) imaging was performed. The subsequent analysis revealed that the elemental composition of Cd, Se, Zn, and S in CdSe/ZnS QDs is approximately 39.42%, 6.86%, 11.66%, and 42.06%, respectively. The results have been added in the revised manuscript.
Comments 2: The authors should provide a clear explanation for the absence of a description of the PMMA grade, molecular weight, or purity. If these details are not necessary for this study, a comprehensive justification from the authors would be greatly appreciated.
Response:
Thanks for the reviewer’s comment. All chemicals used in this study were of analytical grade, the molecular weight of PMMA was 12000.
Comments 3: Are the CdSe/ZnS QDs uniformly dispersed in the composite material? How did the authors confirm this? If there is anything that can be considered from the optical properties, it should be described and discussed.
Response:
Thanks for the reviewer’s comment. Since the end-face color of the CdSe/ZnS QDs doped PMMA fiber appeared uniform under the microscope, and both side and end-face observations under pumping conditions exhibited relatively uniform and good luminescent properties, as seen in Figure 9, we assumed that the QDs were uniformly distributed within the fiber. We sincerely apologize for not conducting an in-depth microscopic examination of the QDs distribution uniformity within the fiber. We will pay special attention to this point in our future work. Thanks and apologies again.
Comments 4: There are many previous studies on the optical properties of CdSe/ZnS/polymer composites, and the authors' study should be compared with these studies to discuss what features are unique to these materials.
Response:
Thanks for the reviewer’s comment. In previous studies, researchers primarily focused on exploring various QDs and polymer materials to achieve improved luminescence properties. The basic methods used for fiber preparation were typically the hollow fiber filling method and the melt-drawing method. However, these methods have certain limitations and are not suitable for producing longer lengths of optical fibers. Therefore, the focus of this paper is not solely on the optical properties of QDs doped polymer fibers but rather on exploring a simple and scalable manufacturing process for fiber production. The discussion and comparison of QDs-doped PMMA fiber preparation methods have been added to the reviewed manuscript in Section 4.1.4.
Comments 5: "2.2.1. hollow fiber filling method": The beginning of the sentence should be capitalized.
Response:
Thanks for the reviewer’s comment. We have corrected this typographical error in the revised manuscript.
Reviewer 2 Report
Comments and Suggestions for Authors
Please, see suggestions on the attached manuscript.
What is the need to include two preparation methods for fibers that are not useful?
Please, add a discussion section.
Finally, homogenize the size of legends in all figures.

Minor editing of English language required.
Author Response
Comments 1: Please, see suggestions on the attached manuscript.
Response:
Thanks for the reviewer’s comment. The scale bar has been added to Figures 7, 8, and 9. The explanation for Figure 10(c) and the unit for the horizontal axis in Figures 10(a) and 10(b) have been clarified in the revised manuscript. The horizontal axis label in Figure 13 has been corrected from "um" to the accurate unit "nm".
The reviewer noted unfamiliarity with the signal strength unit "dBm" used in the manuscript. To clarify, dBm is a unit of power referenced to 1 milliwatt (mW), commonly used to express signal strength or power levels. The calculation is based on the formula: Power (dBm) = 10 × log10 (Power (mW) / 1 mW). This unit is widely used in optical and communications related literature.
Comments 2: What is the need to include two preparation methods for fibers that are not useful? Please, add a discussion section.
Response:
Thanks for the reviewer’s comment. In accordance with the reviewer’s suggestion, Section 4.1.4 has been added to the manuscript to discuss and analyze the necessity of including two preparation methods for fibers that are not useful.
Comments 3: Finally, homogenize the size of legends in all figures.
Response:
Thanks for the reviewer’s comment. The size of legends in all figures have been homogenized.
Reviewer 3 Report
Comments and Suggestions for Authors
The manuscript "Research on CdSe/ZnS Quantum Dots-Doped Polymer Fibers and Their Gain Characteristics" is very well prepared, presenting research that would interest the journal's readers. Therefore, I have very few recommendations for the esteemed authors:
1) In the preparation of the manuscript, the journal template was not used, which is why the line numbers are not visible.
2) Figure 2 is not very legible, so please increase its quality (resolution, contrast, etc.).
3) When publishing an article on an undoubtedly qualitative study like this, it would be appropriate in the Results and Discussion section to have a comparative analysis of the obtained results and dependencies with those of similar studies in the field. Therefore, I ask that such a comparative analysis be added.
The references cited are appropriate.
Author Response
The manuscript "Research on CdSe/ZnS Quantum Dots-Doped Polymer Fibers and Their Gain Characteristics" is very well prepared, presenting research that would interest the journal's readers. Therefore, I have very few recommendations for the esteemed authors:
Comments 1: In the preparation of the manuscript, the journal template was not used, which is why the line numbers are not visible.
Response:
Thanks for the reviewer’s comment, We will pay special attention to this point in our future work.
Comments 2: Figure 2 is not very legible, so please increase its quality (resolution, contrast, etc.).
Response:
Thanks for the reviewer’s comment. We have enhanced the quality of Figure 2 and updated it in the revised manuscript.
Comments 3: When publishing an article on an undoubtedly qualitative study like this, it would be appropriate in the Results and Discussion section to have a comparative analysis of the obtained results and dependencies with those of similar studies in the field. Therefore, I ask that such a comparative analysis be added.
Response:
Thanks for the reviewer’s comment. In response, we have incorporated a comparative analysis of the results in the revised manuscript and have added the relevant references to support this discussion.
Round 2
Reviewer 2 Report
Comments and Suggestions for Authors
Thank you, for considering the suggestions and corrections to enhance manuscript.